# Primordial nucleosynthesis in the era of cosmological tensions

**Cyril Pitrou**[1⋆]

**1** Institut d'Astrophysique de Paris, UMR-7095 du CNRS et de Sorbonne Université, Paris, France

⋆ pitrou@iap.fr ,

## Abstract

**In the few hundreds of seconds of our Universe, the first light nuclei have been synthetized. Starting from an equal amount of neutrons and protons a split second after the big bang singularity, only 12% of baryons are in the form of neutrons when deuterium is produced. A small network of nuclear reactions convert them into approximately 24% of helium in mass fraction. The observational confirmation of this primordial abundance was paramount in establishing the big bang scenario. As these predictions depend crucially on the energy content of the Universe, they are also used to place tight constraints on several variations of the cosmological model.**

## 1 Introduction

As the neutron is unstable, it was suggested in [1] that all elements, which are built out of neutrons and protons bound together, were synthetized when the Universe was very young. We now know that only the first light elements were produced in the first minutes during the so-called big bang nucleosynthesis (BBN). The most striking fact is that approximately 24% of baryons end up in $^4$He [2], whereas the primordial fractions of other light elements with less favorable binding energies are much smaller. Among them, deuterium is measured with great precision from quasar absorption lines and this provides tight constraints on the cosmological model. These predictions depend on the energy content of our Universe, on the abundance of photons with respect to the baryons, but also on the precise knowledge of nuclear reactions. I present the main ingredients of BBN, following [3,4], but I also recommend the reviews [5–9] where the uncertainty on nuclear rates and on observed primordial abundances is discussed more extensively.

## 2 Plasma thermodynamics

The early Universe is dominated by radiation, a term which encompasses photons and neutrinos at least, but also electrons and positrons before they annihilate. Since baryons, a term which encompasses neutrons and protons in cosmology, are non-relativistic once they have formed out of quarks, their energy density scales as $\rho_{\text{bar}} \propto 1/a^3$ and is completely subdominant before the radiation-matter equality. Cosmological expansion induces the redshifting of

temperatures as $T_{\rm rad} \propto 1/a$. In addition, the Friedmann equation $H^2 \equiv (\dot{a}/a)^2 = 8\pi G\rho_{\rm tot}/3$ then implies with $\rho_{\rm rad} \propto T^4$ that $a \propto \sqrt{t}$ such that $T_{\rm rad}^2 \propto 1/t$. At very early times all species are coupled together by scattering and annihilation reactions and they share the same temperature. Whenever the temperature drops below the mass of a given particle, that is when it becomes non-relativistic, annihilations between particles and antiparticles are not balanced by pair creations and this results in a temporary heating of the remaining relativistic particles and a slight departure from the $1/a$ scaling of their tempature.

However neutrinos interact only with electrons and positrons via reactions mediated by weak interactions and they decouple between 2.3 MeV and 1.6 MeV depending on their flavour [10]. Hence when electrons and positrons annihilate around 0.5 MeV, they essentially only reheat photons but not neutrinos. As $T_\nu$ is conserved but not $T_\gamma$ in the process, the photons end up having a larger temperature. The easiest method to estimate the final ratio $T_\gamma/T_\nu$ is to use the conservation of the total volume entropy of the electromagnetic plasma (the bath of photons, electrons and positrons) $\mathrm{d}(a^3 s)/\mathrm{d}t = 0$ . This conservation is valid whenever for all species of the plasma $\mu/T \ll 1$, where $\mu$ is the chemical potential. This approximation is well satisfied in the early Universe. The volume entropy for each species is obtained from the Gibbs-Duhem relation

$$s = \frac{\rho + P}{T} - \frac{\mu n}{T} \, , \tag{1}$$

where $n$ is the number density, and it becomes negligible for non-relativistic particles. Hence in the assumption of negligible chemical potential, we need only to know $\bar{\rho} \equiv \rho/T^4 = 3P/T^4$ of relativistic species. For photons $\bar{\rho}_\gamma = \pi^2/15$, for electrons of both spins $\bar{\rho}_e = 7/8\bar{\rho}_\gamma$ and similarly for positrons. We deduce that for the electromagnetic plasma $s_{\rm pl}/T_{\rm pl}^3$ is reduced by a factor $1 + 2 \times 7/8 = 11/4$ when the electrons and positrons annihilate. Since this is not the case for $s_\nu/T_\nu^3$ as neutrinos have decoupled, we deduce that $T_\gamma/T_\nu = (11/4)^{1/3}$.

In the era of cosmological precision, the previous picture needs to be refined in several ways. First finite-temperature QED effects must be taken into account in the plasma thermodynamics, since this modifies the energy density and the pressure of the plasma, and thus the volume entropy. Also neutrinos are not fully decoupled when electrons and positrons annihilate, hence some pairs of neutrinos/antineutrinos are produced and this modifies the neutrino temperature. Since this production is out of equilibrium, the resulting neutrino spectrum also departs from a Fermi-Dirac spectrum. Furthermore, the various flavours are not produced in equal amounts and neutrino oscillations must be taken into account when describing this process [11–13].

All these effects are characterized by an effective number of neutrinos defined by

$$\rho_{\rm rad} = \rho_\gamma \left[ 1 + \frac{7}{8} N_{\rm eff}(4/11)^{4/3} \right] \tag{2}$$

which would be exactly 3 for three flavours neutrino in absence of the aforementioned small effects. Taking them all into account leads to the prediction $N_{\rm eff} \simeq 3.044$ in the standard cosmological model [13]. Since an energy density is associated with any additional relativistic species, it can also be described as a modification of $N_{\rm eff}$. As detailed hereafter, the abundance of light elements is sensitive to $\rho_{\rm rad}$, hence to $N_{\rm eff}$ via its effect on the abundance of neutrons.

## 3 Weak-interactions

Neutrons and protons are kept in equilibrium in the primordial plasma thanks to the set of six weak interaction reactions

$$n + \nu \leftrightarrow p + e^- \,, \tag{3a}$$

$$n \leftrightarrow p + e^- + \bar{\nu} \,, \tag{3b}$$

$$n + e^+ \leftrightarrow p + \bar{\nu} \,. \tag{3c}$$

The number of neutrons and protons is not only affected by dilution but also by the strength of these reactions, hence the general form of the proton/neutron number density evolution is

$$\dot{n}_n + 3H n_n = -n_n \Gamma_+ + n_p \Gamma_- \,, \tag{4a}$$

$$\dot{n}_p + 3H n_p = n_n \Gamma_+ - n_p \Gamma_- \,. \tag{4b}$$

Assuming that the mass of nucleons is much larger than the temperature of electrons, positrons or neutrinos, and neglecting distortions in the neutrino spectrum, allows to obtain a simple expression for the rates $\Gamma_\pm$ when using the Fermi theory of weak interactions. We get

$$\Gamma_\pm = K \int_0^\infty p^2 \mathrm{d}p [\chi_\pm(E) + \chi_\pm(-E)] \,, \quad \chi_\pm(E) = (E \mp \Delta)^2 g_\nu(E \mp \Delta) g_e(-E) \,, \tag{5}$$

where the mass gap and the Fermi-Dirac distribution functions are

$$\Delta \equiv m_n - m_p \,, \qquad g_\nu(E) = \frac{1}{e^{E/T_\nu} + 1} \,, \qquad g_e(E) = \frac{1}{e^{E/T_{\mathrm{pl}}} + 1} \,. \tag{6}$$

The preconstant $K$ is related to the neutron lifetime $\tau_n$ constant whose measured value is [14] $\tau_n = 878.4 \pm 0.5 \,\mathrm{s}$. Indeed for $T = 0$ the neutrons still decay into protons from the forward reaction (3b), hence $1/\tau_n = \Gamma_+(T = 0)$. We thus estimate $K$ with

$$K^{-1} = \tau_n \int_0^{p_{\max}} p^2 E_\nu^2 \mathrm{d}p \,, \quad p_{\max} = \sqrt{\Delta^2 - m_e^2} \,, \quad E_\nu = \sqrt{p^2 + m_e^2} - \Delta \,. \tag{7}$$

Alternatively it can be obtained from the Fermi constant $G_F$, but this also requires precise knowledge of the CKM mixing angles and of the axial current coupling of nucleons $g_A$, hence it is common to use the precisely measured value of $\tau_n$ to estimate $K$. The correct determination of the weak rates then requires to take into account the QED effects which correspond to extra virtual photons or soft external photons in the Feynman diagrams of the weak interactions. In addition, the mass of nucleons must be taken into account ($m_p = 938.27 \,\mathrm{MeV}$) since they are not infinitely heavy with respect to the typical temperatures in the MeV era. Refined computations also take into account the slightly non-thermal spectrum of neutrinos which departs from a pure Fermi-Dirac spectrum because of their partial reheating mentioned previously.

When interactions are efficient enough, that is for $\Gamma_\pm/H \gg 1$, these reactions enforce the statistical equilbrium which is

$$\frac{n_n}{n_p} = \frac{\Gamma_-}{\Gamma_+} = e^{-\frac{\Delta}{T}} \,. \tag{8}$$

Since $\Gamma_\pm \propto T^5$ whereas $H \propto T^2$, there is a temperature at which the statistical equilibrium is not enforced and the neutron abundance freezes. Since weak interactions are indeed weak, this takes place early in the Universe history, and the freeze-out temperature is estimated to be around $0.8 \,\mathrm{MeV}$. Due to the residual rates the neutron fraction converges toward $X_n^F \simeq 0.16$ around $T_F \simeq 0.3 \,\mathrm{MeV}$. However, neutrons are always converted into protons by beta decay as $\Gamma_+ \to 1/\tau_n$ when $T \to 0$. Hence at lower temperature this temperature is approximately reduced by an extra factor $\exp[-(t - t_F)/\tau_n]$, where $t_F$ is the cosmic time corresponding to $T_F$, as can be checked on the left panel of Fig. 1.

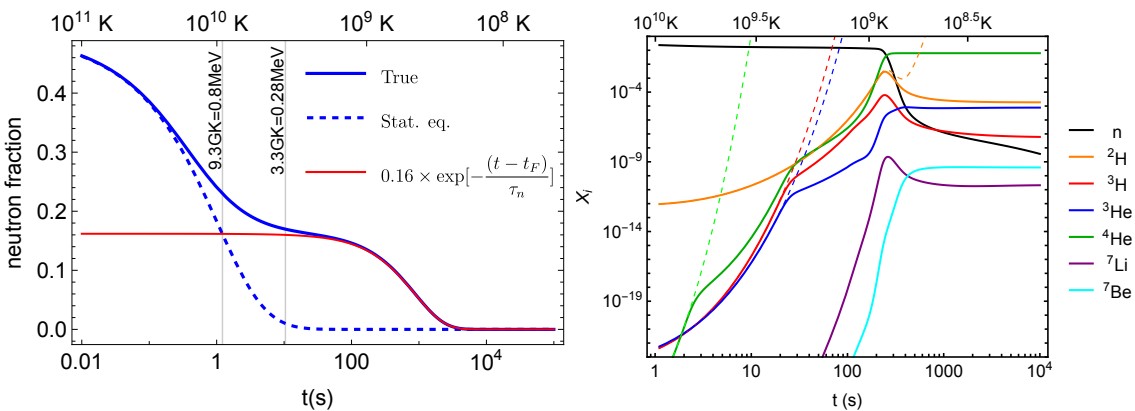

Figure 1: *Left:* Evolution of the neutron fraction with only weak interactions which gradually departs from the statistical equilibrium value. *Right:* Evolution of abundances $X_i$ of the lightest nuclides due to nuclear reactions. The dashed lines correspond to the NSE used for setting initial conditions.

# 4 Nuclear reactions

The typical nuclear reactions are given as evolution of the fraction of a given element in the baryons, that is as equations for $X_i \equiv n_i/n_b$. The evolution of a given fraction depends on the reactions which create and deplete it. Taking only into account two-body reactions of different species, this is of the form (for more general reactions see [4])

$$\dot{X}_i = \sum_{jkl} \Gamma_{kl \to ij} X_k X_l - \Gamma_{ij \to kl} X_i X_j \,. \tag{9}$$

Obtaining primordial nucleosynthesis predictions consists in integrating this set of coupled ordinary differential equations with the weak interaction rates (4) which interchange neutrons and protons, along with the evolution of the electromagnetic plasma temperature (from entropy conservation) and of the scale factor (from Friedmann equation). Popular codes that achieve this are `Parthenope` [15], `AlterBBN` [16] and `PRIMAT` [4].

The rate of a two-body reaction depends on the temperature since is related to the cross-section via

$$\Gamma_{ij \to kl}(T) = n_b \langle \sigma_{ij \to kl}(v) v \rangle_T \,, \tag{10}$$

where the average is over the distribution of relative velocity $v$ of the initial nuclei at a given temperature. The cross section depends in general on the relative velocity and we must rely on measurements performed in laboratory to evaluate the functional dependence. The relative velocity follows a Maxwellian distribution (with the reduced mass of the two-body problem) since all nuclei are non-relativistic, hence it is straightforward to perform the average in (10) once the cross section is known.

The abundances can be initialized with the nuclear statistical equilibrium (NSE) solutions which stem from the fact that for such equilibrium the chemical potential of a any nucleus must be the sum of the chemical potentials of its constituents. For a nucleus $i$, it is

$$n_i^{\text{NSE}} = \frac{g_i m_i^{3/2}}{2^{A_i}} \left(\frac{n_p}{m_p^{3/2}}\right)^{Z_i} \left(\frac{n_n}{m_n^{3/2}}\right)^{A_i - Z_i} \left(\frac{2\pi}{T}\right)^{\frac{3(A_i-1)}{2}} e^{B_i/T} \,, \tag{11}$$

where $m_i$ is its mass, $g_i$ is its spin multiplicity, $A_i$ is its total number of nucleons, $Z_i$ is its proton number and $B_i$ is its binding energy. The abundances will depart from the NSE even

though the thermal equilibrium will always be sastisfied in the early Universe, as momentum transferring collisions are much more efficients than nuclear reactions.

The first reaction which can take place is the production of deuterium

$$n + p \longleftrightarrow D + \gamma \,. \tag{12}$$

However, the forward reaction (synthesis) is only possible below a certain temperature since photons which are too energetic would destroy the newly formed nuclei via the reverse reaction (deuterium dissociation) for large temperatures. There are many more photons than baryons since the baryon to photon ratio, which is conserved after electron/positron annihilation, is

$$\eta \equiv \frac{n_b}{n_\gamma} \simeq 6.1 \times 10^{-10} \times \left( \frac{\Omega_b h^2}{0.02225} \right) \left( \frac{2.7225 \,\mathrm{K}}{T_{\mathrm{CMB}}} \right)^3 \,. \tag{13}$$

As a consequence, the temperature must drop well below the binding energy for the deuterium synthesis to start. We can estimate the temperature at which deuterium is produced in non-negligible amounts. First, the Saha equilibrium for deuterium, which is nothing but the NSE (11), leads for deuterium (when neglecting order one numerical factors) to

$$n_D \propto \frac{n_n n_p}{(m_b T)^{3/2}} e^{B_D/T} \,, \tag{14}$$

where $B_D \simeq 2.2 \,\mathrm{MeV}$ is the deuterium binding energy. Assuming that the condition $n_b \simeq n_p \simeq n_D$ describes the deuterium production condition, we can also approximate these quantities by $\eta n_\gamma$ hence by $\eta T^3$ up to unimportant order one numerical factors. Hence we find that deuterium is produced for a temperature $T_{\mathrm{nuc}}$ which satisfies

$$e^{-B_D/T_{\mathrm{nuc}}} \simeq \eta \left( \frac{T_{\mathrm{nuc}}}{m_b} \right)^{3/2} \simeq 6 \times 10^{-10} \times 10^{-6} \tag{15}$$

where we used $m_b/T_{\mathrm{nuc}} \simeq 10^4$ and (13). This leads to $T_{\mathrm{nuc}} \simeq B_D/35 \simeq 0.65 \,\mathrm{MeV}$. It can be checked on the right panel of Fig. 1 that deuterium is indeed maximal around that temperature.

When deuterium is produced, it is used in a network of nuclear reactions, mostly via the three reactions

$$D + p \longleftrightarrow {}^3\mathrm{He} + \gamma \,, \tag{16a}$$

$$D + D \longleftrightarrow {}^3\mathrm{H} + p \,, \tag{16b}$$

$$D + D \longleftrightarrow {}^3\mathrm{He} + n \,. \tag{16c}$$

The cross-section of the first reaction is now well measured [17] and the uncertainty in BBN predictions is now dominated by the uncertainty in the rates of the last two reactions [18–20].

Other nuclear reactions then convert the products of these reactions into ${}^4\mathrm{He}$, hence almost all neutrons available end up in helium, and only a trace amount of deuterium remains when reactions (16) become inefficient due to cooling and dilution by cosmological expansion. The amount of neutrons available when nuclear reactions take place depends crucially on the time elapsed between freeze-out and the onset of the synthesis. As $T \propto 1/a$ this depends on the Friedmann equation and on $\rho_{\mathrm{rad}}$. The larger the radiation density, the faster it cools and thus the more neutrons are available as less neutrons have been converted to proton via beta decay. This is the main reason why BBN is sensitive to $N_{\mathrm{eff}}$ and this allows to place tight constraints on the number of relativistic degrees of freedom in the primordial Universe.

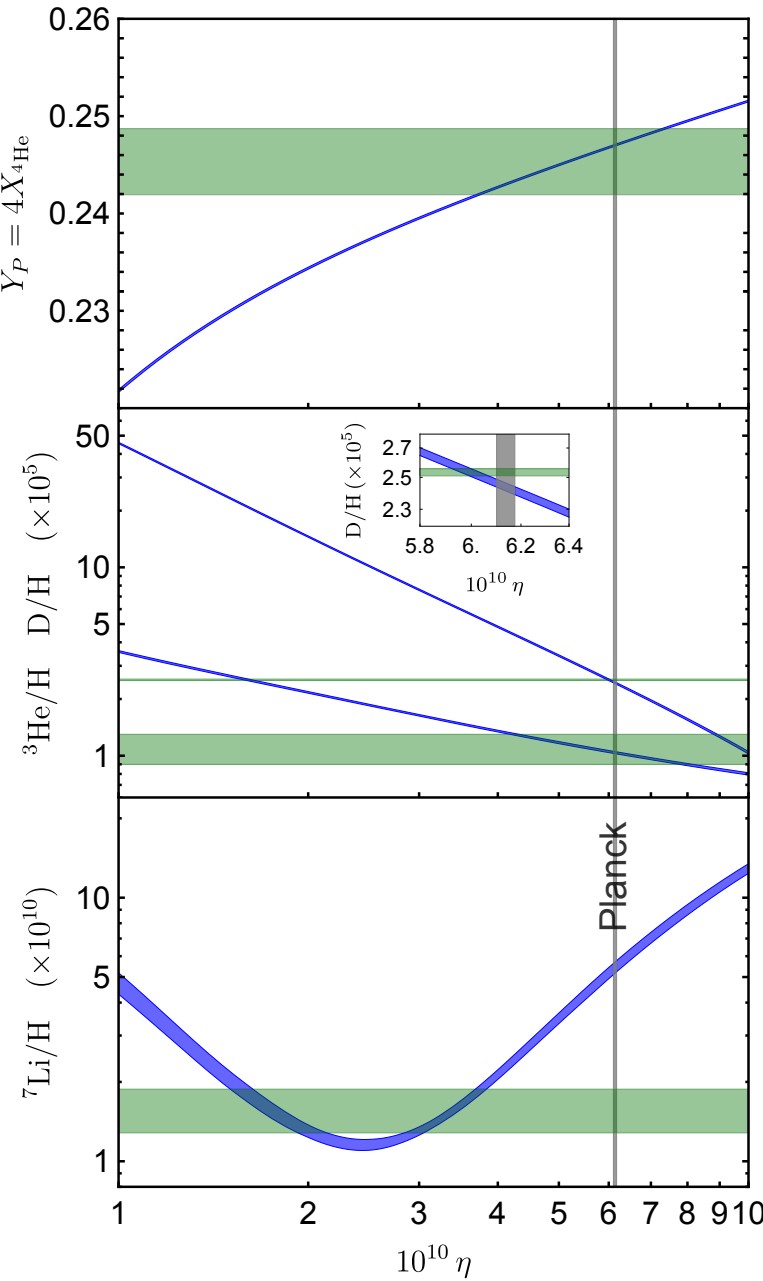

Figure 2: *Green :* observational constraints on light elements. *Gray :* measured value of $\eta = n_b/n_\gamma$ from CMB anisotropies by the Planck satellite [22]. *Blue :* theoretical predictions of BBN with uncertainties from nuclear and weak rates. The BBN estimations of $\eta$ correspond to the regions where the blue and green curves overlap.

# 5 Observational status

For a fixed number of neutrino generations, the final abundances of light elements depend only on the baryon to photon ratio $\eta$. Hence, each light element abundance determination provides in principle an independent measurement of $\eta$. However there are limitations to this method. First, tritium decays into $^3$He hence it is customary to refer to the latter abundance as their sum. Similarly $^3$Li stands for its sum with $^7$Be as the latter decays into the former. Moreover $^3$He is both produced and destroyed in stars and it is not possible to accurately

measure its primordial abundance. The abundance of lithium is indirectly constrained via the observation of the surface of very old stars but it crucially depends on the assumption that it is not destroyed in sizable amounts.

Hence only the abundances of $^4$He and deuterium, which are both very well measured, can be used to confront the BBN theory with observations. The current observational limits are from [2, 21] for deuterium and [2] for $^4$He. The values are

$$Y_P \equiv 4X_{^4\text{He}} = 0.2453 \pm 0.0034, \qquad X_{\text{D}}/X_{\text{H}} \times 10^5 = 2.533 \pm 0.024. \qquad (17)$$

Since $\eta$ is also very well constrained from the baryon acoustic oscillations of the CMB, BBN can also be used as a check of the consistency of the cosmological model. On Fig. 2 we see that the values of $\eta$ obtained from abundances of $^4$He and deuterium are both compatible with the value constrained from CMB alone. Note that there is a slight incompatibility between the value inferred from deuterium and the one from CMB [18] but this remains below the $2 - \sigma$ threshold.

# 6 Conclusion

Even though BBN is as old as the hot big bang model [1], it is still relevant to constrain various extensions of the standard cosmological model. Indeed, since the neutron fraction at the temperature of the synthesis crucially depends on $N_{\text{eff}}$, the aforementioned agreement in the determination of $\eta$ disappears whenever the value of $N_{\text{eff}}$ departs too much from the standard value. All alternative cosmological scenarios with extra relativistic degrees of freedom are thus severely constrained by BBN. This also calls for a better determination of the nuclear rates needed in the theoretical predictions of primordial abundances, especially the last two in (16), so as to match the increasing precision of the observational determination of deuterium.

# Acknowledgements

I am indebted toward Elisabeth Vangioni, Alain Coc, Jean-Philippe Uzan and Julien Froustey.

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
