# Peer review of "Primordial nucleosynthesis in the era of cosmological tensions"

_SciPost Physics Lecture Notes_

## Round 1 · Referee Report · Anonymous (Referee 1) · 2025-9-3

Strengths

1- The author provides a very systematic and clear summary of the complex topic of Big Bang Nucleosynthesis (BBN). The article logically progresses from the thermodynamic background of the early universe to the key weak interaction processes that determine the neutron-to-proton ratio, and finally to the nuclear reaction network that forms the light elements. The entire logical chain is very smooth and reader-friendly. 2- It successfully captures the core physics of BBN, such as the neutron "freeze-out" process, the deuterium bottleneck, and the sensitivity of the final elemental abundances to key cosmological parameters like the baryon-to-photon ratio and the effective number of neutrino species . These are the most important takeaways in this field.

Weaknesses

1- As a lecture note, the target audience likely includes graduate students and researchers from other fields. Some key equations, such as the integral expression for the weak interaction rates, are presented without a brief explanation of their physical origin. This could be confusing for newcomers to the subject.

Report

1- This manuscript provides an excellent, concise, and accurate review of the fundamental principles, calculation procedures, and observational tests of BBN. The author presents a clear narrative, guiding the reader through the key physical steps of how the first light elements were formed in the first few seconds to minutes after the Big Bang. The article not only correctly presents the framework of the standard BBN theory but also includes discussions on high-precision corrections like neutrino decoupling and finite-temperature QED effects. By integrating the latest experimental data, it highlights the remarkable consistency between theory and observation, which stands as a powerful pillar of the standard cosmological model. 2- Overall, this manuscript is a perfect fit for the SciPost Physics Lecture Notes series. It is systematic, accurate, easy to understand, and covers a topic of ongoing interest to the cosmology and particle physics communities. It serves as an excellent introductory material for students or researchers entering the field.

Requested changes

1- It would be beneficial to add a sentence or two around key physical equations (such as Equations 5 and 11) to briefly explain the physical origin or meaning of the terms. This would significantly enhance the pedagogical value of the lecture note, making the content more accessible to non-specialist readers. 2- Fig. 2 is one of the core figures of the article, and its inset is very useful. It is suggested to slightly increase the font size of the axis labels and numbers for both the main plot and the inset to improve readability.

Recommendation

Ask for minor revision

---

## Round 1 · Referee Report · Anonymous (Referee 2) · 2025-10-1

Disclosure of Generative AI use

The referee discloses that the following generative AI tools have been used in the preparation of this report:

I used Gemini to verify the argument of 3Li should be corrected to 7Li, I noticed the mistake but I wanted to be sure.
I also used it to help me correct for punctuation and grammar in my report.

Strengths

The lecture notes are clearly structured and written in an accessible style, making them suitable for advanced students and researchers wishing to familiarise themselves with BBN. The derivations are pedagogical, and the connection to cosmological tensions makes the material timely and relevant. Additionally, these notes are shorter in comparison to others, but provide a direct pedagogical connection that can is very useful.

Weaknesses

On the Tritium Decay Chain.
The text states that $\text{Tritium}$ (${^3H}$) decays into ${^3He}$. This is chemically correct, but maybe for a lecture note intended to be pedagogical, it would be beneficial to add a note that the ${^7Be}$ decay into ${^7Li}$ occurs specifically via electron capture. This detail adds completeness, especially as the ${^7Li}$ abundance remains the most notable discrepancy between BBN theory and observation.

Target audience.
Since the lecture notes are tailored to people who are not experts in the field, some equations and approximations (e.g. neglecting chemical potentials, using order-one factors in deuterium equilibrium estimates) are made quickly. A short explanatory sentence in each case would make the text more accessible for students and non-experts.

Report

The manuscript presents a pedagogical introduction to big bang nucleosynthesis (BBN), covering plasma thermodynamics, weak interactions, nuclear reactions, and observational constraints. The author provides a concise but thorough overview of the theoretical framework and links it to current cosmological tensions, particularly those related to light element abundances and the effective number of neutrino species.

I recommend acceptance after minor revisions. The manuscript is clear, correct, and well suited for publication in SciPost Lecture Notes, but I encourage the author to expand slightly on the discussion of cosmological tensions, and to consider adding one or two pedagogical figures (e.g., a BBN reaction network diagram).

Requested changes

Minor corrections:
``sastisfied'' $\to$ ``satisfied'' (p. 5).
``equilbrium'' $\to$ ``equilibrium'' (p. 3).
On p. 6, ``$^3$Li'' is meant to be $^7$Li or ``lithium-7''???
Eq.~(2): briefly remind the reader that $\rho_\gamma$ is the photon energy density.
Eq.~(15): the approximation step could be explained more clearly (mention dropping order-one factors).
Eq.~(17): clarify that $Y_P = 4X_{^4\mathrm{He}}$ corresponds to the helium mass fraction.
Fig.~1: axis labels are a bit small compared to the text; increasing font size would improve readability.
The phrase ``up to unimportant order one numerical factors'' could be rephrased as ``up to factors of order unity, which do not significantly affect the estimate''.

Attachment

Recommendation

Ask for minor revision

---

## Round 1 · Referee Report · Anonymous (Referee 3) · 2025-10-12

Strengths

1- An extremely clear presentation of BBN. Very concise and excellently explained. 2- I really like the explanation of the importance of $N_\rm{eff}$ in the formation of the light elements and the extent to which it is constrained by data.

Weaknesses

1- Needs correcting for some spelling and grammar mistakes. 2- Figures could be improved by being more consistent with each other (larger fonts, labels, right-hand plot of Fig.1 in particular). 3- I think the lectures would benefit from more explanation in the observational status section.

Report

Overall the notes are of excellent quality and clearly recommended for publication. Thank you for giving such a nice lecture, I really enjoyed it!

Requested changes

1- Spelling and grammar corrections: - Synthetized -> Synthesized (abstract, p1) - big bang -> Big Bang (abstract, p1) - 'I also recommend the reviews [5-9] ' should have a comma after the references - tempature -> temperature (p.2) - Commas after Eq.(2) and Eq.(15) - equilbrium -> equilibrium (above Eq.(8) on p. 3) - sastisfied -> satisfied (top of p. 5) - efficients ->  efficient (top of p.5) - Flavour -> Flavor if using US spelling (which was used elsewhere) (3 times on p.2) - 'The mass of nucleons must be taken into account (m_p=...)' -> 'The mass of nucleons must be taken into account (m_p=...),' (should have a comma) - When referring to equations they should be ' in Eq.(1)' ... Not 'in (1)'

2- Typo at bottom of page 6. It currently reads ${}^3\rm{Li}$ but should be ${}^7\rm{Li}$.

3- In Figure 1: Make the labels and legend in the right-hand plot larger. The formatting across the two plots could be more consistent in general. It looks like Latex wasn't used in the labels (in comparison to Fig.2).

4- In Figure 2: The Label for the Planck constraints could be better placed to avoid the line passing through it. The tick marks could be larger.

5- Above Eq.(2) it would be good to be clear that $N_\rm{eff}$ is the effective number of neutrinos: 'All these effects are characterized by an effective number of neutrinos, $N_\rm{eff}$, defined by'.

6- I think you should mention in more detail the Lithium problem. I assume this is what you are touching upon with the sentence 'The abundance of lithium is indirectly constrained via the observation of the surface of very old stars but it crucially depends on the assumption that it is not destroyed in sizeable amounts.' Maybe refer to the figure after this?

7- In Figure 2: make it clear that we are looking at the abundance of the light elements ie abundance of ${}^7\rm{Li}$ with respect to H for example. eg 'observational constraints on the abundance of light elements as a fraction of the abundance of hydrogen. In the top panel we plot the helium fraction defined in Eq.(17).'

8- In Eq.(17) it would be nice to be reminded of what $X_{{}^{4}\rm{He}}$ means. ', where $X_{{}^{4}\rm{He}}=n_{{}^{4}\rm{He}}/n_b$' .

9- How ${}^{4}\rm{He}$ and deuterium are measured could be mentioned in passing in the observational status section (as was mentioned in the abstract eg quasar absorption lines etc).

10- It might be nice to include a sentence or two on how the CMB is used to provide observational evidence.

Recommendation

Ask for minor revision

---

## Editorial Decision

in_refereeing